# OpenReview forum: "Local Superior Soups: A Catalyst for Model Merging in Cross-Silo Federated Learning"
_NeurIPS.cc/2024/Conference — NeurIPS 2024 poster_

### Official Review · Reviewer_9KZz · 2024-06-25

**Soundness:** 3
**Presentation:** 3
**Contribution:** 3
**Rating:** 5
**Confidence:** 3

**Summary:**

In this paper Local Superior Soups (LSS) is proposed to minimize communication rounds in federated learning (FL) using pre-trained models, specifically tackling data heterogeneity challenges. LSS achieves this by employing sequential model interpolation, maintaining connectivity, and integrating diversity and affinity regularization terms. These innovations enable more local training steps and fewer communication rounds, effectively preventing client drift. Designed for adapting pre-trained models in FL, LSS enhances training efficiency, making it well-suited for deployment in edge computing applications.

**Strengths:**

1. The proposed method can effectively reduce the communication rounds in federated learning (FL) using pre-trained models.

2. The proposed method seems sound.

3. This paper is well written.

**Weaknesses:**

1. Only two small-scale image datasets are used in experiments. More large-scale datasets, especially those in other modalities, should be used.

2. More pretrained models should be explored.

3. More tasks besides image classification should be incorporated into experiments.

**Questions:**

None.

**Limitations:**

See weakness part

---

> ### Author Rebuttal · Authors · 2024-08-07
>
> Thank you for your constructive feedback on our manuscript. We appreciate your positive comments on the effectiveness and soundness of our proposed method, as well as the clarity of our writing.
>
> **Expermental Scope Clarification.**
> We would like to clarify that the current set of experiments is sufficient to support our claims and demonstrate the effectiveness and generalizability of our proposed method. Additionally, other reviewers have acknowledged the comprehensiveness of our experiments. Reviewer 2F5f noted, **"The proposed algorithm is tested on a variety of datasets and types of distribution shifts,"** and Reviewer 3kGS commented, **"This paper conducts extensive experiments to illustrate the effectiveness of LSS."**
>
> However, we also recognize that extending our experiments could further enhance our experimental scope, particularly with initial explorations on large language models (LLMs).
>
>  We have addressed your concerns as follows:
>
> **1. Issue: Additional Evaluations on Large-Scale Models besides Image Classification**
>
> **Action Taken.**
> We evaluated our proposed method, LLSS, against FedAvg on a large language model for a multilingual instruction tuning task to demonstrate the generalizability of LSS. Additional details can be found in the Appendix.
>
> **Experiment Detail.**
> Evaluation of Large Language Models for Multilingual Instruction Tuning.
> **Setup.** We follow the setup of Fed-Aya [1], which involves four iterative steps as common federated learning: server-to-client model downloading, local model training, client-to-server model uploading, and global model aggregation. For instruction tuning, we use the parameter-efficient fine-tuning technique, LoRA applied to the Llama2-7b model.
>
> **Dataset.**
> We use the Aya dataset [1], a multilingual instruction tuning dataset with annotations from contributors worldwide. Our experiments include 6 high-resource languages (English, Spanish, French, Russian, Portuguese, Chinese) and 2 low-resource languages (standard Arabic, Telugu). The dataset is filtered to include contributors with at least 100 annotations, resulting in 38 clients with a total of 25k data samples.
>
> **Model.**
> The model used for our experiments is the Llama2-7b, fine-tuned using the LoRA technique. We evaluate the effectiveness of the training methods using an in-domain evaluation metric termed Ref-GPT4, where GPT-4o rates the generated responses against ground-truth responses. The score given by GPT-Ref ranges from 0 to 10. We adopt the same prompt template used in FedLLM-Bench [1]. The implementation of applying our method to LoRA is the same as that used in the ViT experiments (see Fig.4 described earlier).
>
> **Result.**
> **In Figure 9 in the Author Rebuttal Attached File**, our method, LSS, when applied to large language models for instruction tuning, achieves higher scores than the common FedAvg. This suggests that LSS is a promising approach for improving performance and convergence in federated learning settings for large language models, in addition to its success in image classification. Exploring the use of our method in a diverse set of complex LLM tasks is an interesting direction for future research.
>
> [1] Ye, Rui, et al. "FedLLM-Bench: Realistic Benchmarks for Federated Learning of Large Language Models." arXiv preprint arXiv:2406.04845 (2024).
>
>
> **2.  Issue: Additional Evaluations on More Pre-trained Models**
>
> **Clarification**
> We have added evaluations on various pre-trained models, including ResNet18, ResNet50, ViT, and LLMs (llama2-7b). We believe our experimental evaluation covers a diverse range of model architectures.

---

> ### Author Response · Authors · 2024-08-13
>
> We sincerely appreciate the time and effort you have invested in reviewing our work and providing such valuable feedback. We hope that our rebuttal has thoroughly addressed your comments and concerns.
>
> As the author-reviewer discussion period nears its conclusion on August 13th, we would like to kindly invite you to contact us if there are any additional questions or points that you believe require further discussion. Your insights have been crucial in refining our submission, and we remain fully committed to addressing any remaining issues.
>
> If you find that our responses and improvements adequately address your concerns, we would be deeply grateful if you could consider raising your score. Thank you once again for your thoughtful review.

---

### Official Review · Reviewer_3kGS · 2024-07-08

**Soundness:** 3
**Presentation:** 3
**Contribution:** 2
**Rating:** 5
**Confidence:** 4

**Summary:**

This paper proposes a method called Local Superior Soups (LSS), a novel technique for model merging in cross-silo federated learning aimed at reducing communication rounds while enhancing model performance. This paper introduces random interpolation, diversity term, and affinity term to alleviate the need for time-consuming model selection and redundant model training. Rigorous experiments on 4 datasets with 11baselines demonstrate the effectiveness of LSS.

**Strengths:**

1. This paper discusses the importance of bridging two low-loss valley to reduce communication rounds.
2. This paper introduces two quantifiable metrics, diversity and affinity, which serve as indicators of model quality..
3. This paper conducts extensive experiments to illustrate the effectiveness of LSS.

**Weaknesses:**

1. The distinction between LSS and similar federated learning methods such as FedProx, which also incorporates weights from global models to regulate client loss, is not clearly discussed.
2. The subsection 3.3.1 titled "Random interpolation conserving connected low-loss region." lacks mathematical detail to fully understand the interpolation process.
3. The requirement for clients to receive the interpolated model pool ($M$) could potentially lead to significant communication overheads, which may not present a clear advantage over simpler methods like FedAvg or FedProx.
4. The connection between Theorem 3.1 and the core methodology of LSS, specifically the diversity and affinity terms, appears tenuous. These terms do not seem to be directly derived from the theorem, which may weaken the theoretical foundation of the proposed method.
5. This paper should consider referencing relevant literatures or conducting preliminary experiments to support its statments on the part called "Limitation of previous model soups methods".

**Questions:**

1. In Figure 3, how do the results of LSS compare with those of FedProx? Given the similarities in the core concepts between LSS and FedProx, such a comparison would be insightful for evaluating the distinct advantages of LSS.
2. The results presented in Tables 1 and 2 suggest that LSS performs well in the initial training rounds. Can the authors clarify whether LSS is primarily advantageous only during these initial rounds?
3. Furthermore, how should the training process be continued post-initialization? Is it feasible to employ LSS throughout the entirety of the training process, or would alternative methods be more effective in later stages?

**Limitations:**

Please see the weaknesses and the questions.

---

> ### Author Rebuttal · Authors · 2024-08-07
>
> We appreciate your thorough review and valuable feedback on our paper. Thank you for your positive comments on the comprehensiveness of our experiments and for recognizing the key contribution of our proposed two metrics (i.e., affinity and diversity) in quantifying model quality for model selection in model interpolation.
>
> Below are our responses to the key points raised:
>
> **1. Comparison with Similar Methods**
>
> **Response.**
> We have discussed the differences between our method and the most relevant existing methods (model averaging) in the Method and Experiment section (see **Line 202 to 220** and **Line 300 to 302**). Our proposed LSS consists of three novel key components: random model interpolation, diversity regularization, and affinity regularization, which is fundamentally different from FedProx.
>
> **Clarification.**
> Compared with FedProx, our LSS contains random model interpolation and diversity regularization two novel components that help reduce the communication cost, the only similarity between LSS and FedProx is the presence of a regularization term: ours leverages affinity regularization, while FedProx uses proximal regularization. However, they **differ significantly in their motivation, implementation, and focus**.
>
> (a) **Motivation**: The affinity term in LSS is designed to ensure that the expansion of low-loss regions remains close to the shared initialization point when incorporating diversity regularization. This increases the likelihood of overlapping connected regions (see **Line 81 to 91**), thereby enhancing performance and convergence by leveraging pre-trained models and significantly reducing communication rounds. In contrast, FedProx aims to improve general convergence and performance in the presence of statistical heterogeneity.
>
> (b) **Implementation**: In LSS, the affinity term minimizes the distance of local models from the **initialized model parameters** (i.e., the pre-trained model parameters), **rather than the global model** (see **Line 81 to 83**, **Algorithm 1 Line 2** and Require Description). In contrast, FedProx directly incorporates global model weights to regulate client loss.
>
> (c) **Focus**: FedProx primarily introduces a regularization term to address general convergence and performance issues arising from statistical heterogeneity. In contrast, our method focuses on leveraging pre-trained models to achieve significant communication round savings. We approach this from a novel perspective, using model interpolation techniques to enhance FL convergence and performance. Additionally, we introduce an affinity regularization term to improve model quality during interpolation. Our **affinity term is intrinsically linked to the diversity term and model interpolation process** (see **Line 81 to 94**, and **Line 252 to 255**), making it an **integral and non-isolated part of our approach**.
>
> **Added Experiment.**
> To clearly illustrate the distinct advantages of LSS, we have included comparative results with FedProx similar to those in Figure 3 in the **Author Rebuttal Attached File. Fig.8.** This comparison highlights the benefits of our approach in terms of communication efficiency and performance.
>
> **2. Mathematical Detail of Random Interpolation: The subsection 3.3.1 lacks mathematical detail to fully understand the interpolation process.**
>
> **Response:** We will revise subsection 3.3.1 to provide a more detailed mathematical explanation of the random interpolation process.
>
> **3. Communication Overhead: Potential significant communication overhead due to the requirement for clients to receive the interpolated model pool.**
>
> **Clarification:** We would like to clarify that our LSS method does not lead to additional communication overhead. Since we use the same model interpolation operation after model selection as model soups, **the local model is merged from the interpolated model pool before communication**. This ensures that our method maintains the same level of overhead as basic methods like FedAvg. In our revised version, we will introduce more background on model soups in Section 1 to improve clarity regarding communication overhead in parameters.
>
> **4. Theoretical Connection**
>
> **Response.** Theorem 3.1 aims to provide a convergence guarantee when our proposed regularization terms are added, not to derive these terms directly from the theorem. In revision, we will clarify the motivation and role of the regularization terms at the beginning of Theorem 3.1.
>
> **5. Reference and Statement Support**
>
> **Response.** We would like to clarify that we have already included references to support our statements in Section 3 (see **Line 203**). Although the limitation is not explicitly stated in the references, it can be clearly derived from the algorithm process. In revision, we will expand this part by adding more related work to better support our claims.
>
>
> **6. Experiment Scope: The results presented suggest LSS performs well in initial training rounds. It is unclear if LSS is advantageous throughout the training process.**
>
> **Response:** Yes, as indicated in Figure 3, our LSS performs well not only during the initial training rounds but also throughout the entire training process. Additionally, we would like to highlight that LSS is designed to reduce communication rounds when using pre-trained models. Training with 'continued post-initialization' is not the focus of our problem setting.
> Specifically, we aim to emphasize that LSS is tailored to minimize communication rounds in the context of pre-trained models. However, the design of our method should be compatible with existing algorithms for post-initialization training. Examining the effects of post-initialization training falls beyond the scope of our current paper, we will explore this topic in our future research work.
>
>
> We believe these revisions significantly enhance the clarity of our paper. Thank you again for your insightful comments and suggestions.

---

> > ### Comment · Reviewer_3kGS · 2024-08-11
> > **Thank you for your answers.**
> >
> > Thank you for your answers. After reading your clarifications, I have raised my score, and I have no other concerns. I appreciate your effort.

---

> > > ### Author Response · Authors · 2024-08-11
> > > **Appreciation for the Positive Feedback and Score Increase**
> > >
> > > We are truly grateful that our responses have successfully addressed all of your concerns. Thank you for your positive feedback and for raising our score. We deeply appreciate your valuable comments and the time you’ve taken to review our work!

---

### Official Review · Reviewer_2F5f · 2024-07-13

**Soundness:** 3
**Presentation:** 2
**Contribution:** 2
**Rating:** 5
**Confidence:** 4

**Summary:**

This paper proposes LSS, a model interpolation-based local training technique to reduce the number of communication rounds required. The intuition is to regularize local models to connected low-loss valleys, so the aggregated model may have lower loss. LSS is empirically evaluated on a variety of datasets and types of distribution shifts.

**Strengths:**

- Figure 1 and 2, though may not be very rigorous, are clear and provide intuition to the readers.
- The proposed algorithm is tested on a variety of datasets and types of distribution shifts.

**Weaknesses:**

- Readability: the notation in section 3 is not clear enough. For example, $n$ is used for both number of data (section 3.1) and number of averaged model (Alg 1), which is confusing.
- (minor) It might be a little abuse of notation when using $\mathcal{D}_i$ for both distribution and dataset. I suggest using different notations.

**Questions:**

- Although Figure 2 is intuitive, it might not be so rigorous: high affinity and high diversity do not necessarily guarantee an aggregated model with low loss. In the example, two client’s local models are different “horizontally”, while the major axis of the loss landscape is also horizontal. However, this is not always guaranteed. If the clients models are different “vertically”, the proposed method may just fail. Mathematically, in algorithm 1, the gradient of $dist(f_{p_i}, f_p)$ and $dist(f_{p_i}, \mathcal{M})$ is very likely to be nearly orthogonal. Could you explain intuitively why LSS is expected to perform well in most of the cases?
- The scope of this paper is limited to FL with a pre-trained model. I understand that the pre-trained model may be larger in average, which makes convergence more changing. I am curious how LSS is limited to pre-trained models and whether it can generalize to randomly initialized models.

**Limitations:**

Yes.

---

> ### Author Rebuttal · Authors · 2024-08-07
>
> Thank you for your thoughtful and constructive feedback. We appreciate that you found our figure illustrations clear and our experimental scope comprehensive, covering diverse datasets and types of distribution shifts. We also value your suggestions for improving our manuscript.
>
> We have addressed your concerns as follows:
>
> **1. Notation Clarity**
>
> **Revision:**
> - We will revise Algorithm 1, changing the interpolated model number notation from "n" to "N" (Line 4 in Algorithm 1) to distinguish it from the sample number.
> - In Section 3, we will provide different notation and description for distribution and dataset to prevent any potential confusion. Change Line 136 to 137 into
> "For each client, we have access to `$n`$ training data points in the form of `$(\mathcal{X}_i, \mathcal{Y}_i) = \{(x_{j}^{i}, y_{j}^{i})\}_{j=1}^{n}`$, where `$y_{j}^{i}`$ denotes the target label for input `$x_{j}^{i}`$. "
> - We will add a reference note in Section 3.1 directing readers to the notation table in Appendix A for convenient access to corresponding notations and descriptions.""
>
> **2. Question: Could you explain intuitively why LSS is expected to perform well in most of the cases?**
>
> **Response.**
> We acknowledge that high affinity and high diversity do not necessarily guarantee an aggregated model with low loss due to the complexity of the high-dimensional loss landscape. However, our empirical evaluations show that using pre-trained model initialization, combined with our proposed random interpolation and the terms affinity and diversity, positively impacts performance and convergence across various types of distribution shifts.
>
> **Clarification.**
> Our method performs well in most cases when using overparameterized pre-trained models. Typically, training with a pre-trained model results in small initial gradient dissimilarity, and overparameterization ensures that the optimal parameters are close to the initialization. See Section 3 (see **Line 181 to 183**) for more detailed explanations. In summary, **small gradient dissimilarity among clients** and a **small parameter distance from initialization to optima**, combined with leveraging our introduced diversity and affinity terms, make our method work well in most cases when training with pre-trained models.
>
>
> **3. Limitation: The method is tested only with pre-trained models. It is unclear how it would perform with randomly initialized models.**
>
> **Response.** Our LSS approach is also effective in improving performance and convergence when training models with random initialization. However, it does not achieve the impressive few-round near-optimal performance seen when training with pre-trained models.
>
> **Added Experiment.** In response to your query, we have added performance results comparing the training with randomly initialized and pre-trained parameters in the **Author Rebuttal Attached File Table 11**.
>
> Thank you again for your valuable feedback. We believe that these revisions significantly strengthen our manuscript, and we appreciate your feedback in making these improvements.

---

> ### Author Response · Authors · 2024-08-13
>
> We sincerely appreciate the time and effort you have invested in reviewing our work and providing such valuable feedback. We hope that our rebuttal has thoroughly addressed your comments and concerns.
>
> As the author-reviewer discussion period nears its conclusion on August 13th, we would like to kindly invite you to contact us if there are any additional questions or points that you believe require further discussion. Your insights have been crucial in refining our submission, and we remain fully committed to addressing any remaining issues.
>
> If you find that our responses and improvements adequately address your concerns, we would be deeply grateful if you could consider raising your score. Thank you once again for your thoughtful review.

---

### Author Rebuttal · Authors · 2024-08-07

We sincerely thank the AC and reviewers for their valuable time. We are delighted to see that the reviewers highlight our **"paper is well written"**, recognizing the importance of our idea of **"bridging two low-loss valley to reduce communication rounds"**, and our proposed method **"effectively reduce the communication rounds"** and **"tested on a variety of datasets and types of distribution shifts."**

This rebuttal clarifies 2F5f’s concern on the notation usage, 3kGS’s questions regarding our method design and motivation, 9KZz’s concerns about validation on more datasets, and addresses all other minor questions and discussion points from all reviews.

Below is a summary of our rebuttal:

**Response to Reviewer 2F5f:**

1. **Notation Clarity:**
Revised notation in Algorithm 1 and Section 3 for better clarity. Added a reference note to the notation table in Appendix A.

2. **Explanation of LSS Performance:**
Clarified that training with pre-trained models and regularized model interpolation promotes a low-loss connection, enhancing performance.

3. **Limitation of Pre-trained Models:**
Clarified that LSS is effective with random initialization but excels with pre-trained models. Added comparative results in the attached file.

---

**Response to Reviewer 3kGS:**

1. **Comparison with Similar Methods:**
Differentiated LSS from FedProx in terms of implementation, motivation, and focus. Added comparative results in attached file highlighting LSS's advantages.

2. **Mathematical Detail of Random Interpolation:**
Revised subsection 3.3.1, including detailed mathematical explanations and relevant formulas.

3. **Communication Overhead:**
Clarified that LSS maintains similar communication overhead to methods like FedAvg.

4. **Theoretical Connection:**
Clarified the role of diversity and affinity terms in supporting convergence guarantees in Theorem 3.1.

5. **Reference and Statement Support:**
Included additional references to support our claims.

6. **Experiment Scope:**
Clarified LSS's effectiveness throughout the training process.

---

**Response to Reviewer 9KZz:**

1. **Experimental Scope Clarification:**
Added evaluations on a large language model for multilingual instruction tuning. Clarified that various pre-trained models (ResNet18, ResNet50, ViT, LLM) are included to demonstrate the generalizability of LSS.

---

### Decision · Program_Chairs · 2024-09-25

**Decision:**

Accept (poster)

**Comment:**

All three reviewers have provided positive feedback on this submission. The paper introduces a novel method to reduce communication costs in federated learning for pre-trained models. The authors have adequately addressed the concerns and weaknesses raised by the reviewers. Based on this, I recommend acceptance.